# BDNF Expression in Cortical GABAergic Interneurons

**DOI:** 10.3390/ijms21051567

**Published:** 2020-02-25

**Authors:** Federico José Barreda Tomás, Paul Turko, Heike Heilmann, Thorsten Trimbuch, Yuchio Yanagawa, Imre Vida, Agnieszka Münster-Wandowski

**Affiliations:** 1Institute of Integrative Neuroanatomy, Charité-Universitätsmedizin Berlin, Campus Mitte, 10117 Berlin, Germany; federico.barreda@gmail.com (F.J.B.T.); paul.turko@charite.de (P.T.); heike.heilmann@charite.de (H.H.); imre.vida@charite.de (I.V.); 2Bernstein Center for Computational Neuroscience (BCCN) Berlin, 10115 Berlin, Germany; 3Institute of Neurophysiology, Charité - Universitätsmedizin Berlin, Campus Mitte, 10117 Berlin, Germany; thorsten.trimbuch@charite.de; 4Departments of Genetic and Behavioral Neuroscience, Graduate School of Medicine, Gunma University, Graduate School of Medicine, Maebashi City 371-8511, Japan; yuchio@gunma-u.ac.jp

**Keywords:** BDNF, hippocampus, neocortex, GABAergic interneurons, laser capture microdissection, RT-qPCR, FACS

## Abstract

Brain-derived neurotrophic factor (BDNF) is a major neuronal growth factor that is widely expressed in the central nervous system. It is synthesized as a glycosylated precursor protein, (pro)BDNF and post-translationally converted to the mature form, (m)BDNF. BDNF is known to be produced and secreted by cortical glutamatergic principal cells (PCs); however, it remains a question whether it can also be synthesized by other neuron types, in particular, GABAergic interneurons (INs). Therefore, we utilized immunocytochemical labeling and reverse transcription quantitative PCR (RT-qPCR) to investigate the cellular distribution of proBDNF and its RNA in glutamatergic and GABAergic neurons of the mouse cortex. Immunofluorescence labeling revealed that mBDNF, as well as proBDNF, localized to both the neuronal populations in the hippocampus. The precursor proBDNF protein showed a perinuclear distribution pattern, overlapping with the rough endoplasmic reticulum (ER), the site of protein synthesis. RT-qPCR of samples obtained using laser capture microdissection (LCM) or fluorescence-activated cell sorting (FACS) of hippocampal and cortical neurons further demonstrated the abundance of BDNF transcripts in both glutamatergic and GABAergic cells. Thus, our data provide compelling evidence that BDNF can be synthesized by both principal cells and INs of the cortex.

## 1. Introduction

Brain-derived neurotrophic factor (BDNF), a neuron-specific survival factor, was first isolated from the pig brain [1]. It is involved in multiple subcellular pathways and is critical for central nervous system (CNS) development, including neuronal differentiation, survival and synaptic plasticity [1,2,3]. BDNF is considered to be a general modulator of neurotransmitter release, including the release of gamma-Aminobutyric acid GABA [4]. BDNF is initially synthesized as a ~37 kDa glycosylated precursor protein (pro)BDNF, which is post-translationally cleaved to generate the mature ~13 kDa mBDNF protein [5,6]. In rodents, there are 22 BDNF transcripts encoding the same protein: eleven different 5’ untranslated regions (UTRs) are alternatively spliced to a common downstream exon IX containing the coding region [7,8]. These transcripts differ in terms of their spatial and temporal distribution within subcellular compartments, nevertheless producing the same protein which, depending on its spatiotemporal localization, controls different aspects of neuronal function [9,10]. For example, in rat hippocampal neurons, somatically synthesized BDNF has been suggested to promote dendritic spine formation, whereas dendritically synthesized BDNF appears to be a key regulator of spine maturation [11]. BDNF can also promote synaptic plasticity by binding to the high-affinity tyrosine kinase receptor, tropomyosin receptor kinase B (TrkB) [12].

Although BDNF is essential for the development and survival of all neural subpopulations [13,14,15], it has been shown to play a particularly important role in promoting the dendritic development of GABAergic interneurons (INs) [16]. This is notable, given that BDNF is postulated to be produced exclusively by glutamatergic excitatory neurons [17]. Thus, other cell populations have been assumed to rely on the activity-dependent secretion of BDNF by glutamatergic principal neurons for their survival and development [17,18]. Indeed, BDNF protein isoforms in the hippocampus are apparent in principal neurons of cornu ammonis area 1 and 3 (CA1, CA3) and dentate gyrus (DG) as early as postnatal day 4, and expression levels in these cells are more abundant than in the surrounding tissue. The amount of mBDNF protein peaks at around postnatal day 28, before stabilizing at comparatively reduced levels in the adult [19]. Consistently, the expression of BDNF mRNA follows a similar expression pattern [19] and its level also rises from early postnatal ages to adulthood [20,21].

While BDNF RNA transcripts are widely assumed to be absent from cortical inhibitory INs [2,22], recent experimental data hint at the presence of BDNF RNA in these neurons, both during development and in the adults. In the adult hippocampus, a recent in situ hybridization study showed that endogenous BDNF transcripts are primarily localized to neuronal somata residing in the principal cell body layers [23], but they were also observed in cells scattered throughout the dendritic layers, indicative of displaced pyramidal cells or different types of INs [23]. Similarly, in the visual cortex, transgenic BDNF overexpression was predominantly, but not exclusively, restricted to pyramidal neurons [24]. In their study, Huang et al. (1999) observed faint, but compelling, immunoreactivity for BDNF in the somata of a proportion of parvalbumin-positive INs [24]. Aside from cortical expression, there is also evidence that a subpopulation of inhibitory amacrine cells in the retina can be a source of BDNF [25]. BDNF expression has also been observed in GABAergic neurons during neural tube development, with BDNF mRNA clearly present in ventrally projecting axon-bearing INs and at later stages in INs of the dorsal spinal neural tube [26].

Thus, whether or not GABAergic INs are capable of synthesizing BDNF remains unclear. In this study, therefore, we addressed the localization and expression of the mature and precursor BDNF proteins and their RNA transcripts in hippocampal neurons by applying a combination of laser capture microdissection (LCM) and fluorescence-activated cell sorting (FACS), together with real-time reverse transcription quantitative PCR (RT-qPCR) and immunocytochemical analysis. Our results convincingly demonstrated for the first time that the two major cortical neuron classes, glutamatergic principal cells (PCs) and GABAergic INs, can synthesize BDNF.

## 2. Results

### 2.1. Identification of Cortical Excitatory Principal Cells and GABAergic Interneurons in Transgenic Mice

For all immunocytochemical, LCM and FACS analyses, we utilized two transgenic mouse lines expressing fluorescent reporter proteins in either GABAergic neurons only or in both GABAergic and glutamatergic neurons. In order to identify GABAergic INs in the hippocampus and neocortex, we used a vesicular GABA transporter-yellow fluorescent protein (VGAT-YFP) transgenic mouse line selectively expressing the YFP, Venus, under the VGAT promoter in INs [27]. In this mouse line, 97.3% of all hippocampal and 95.9% of neocortical GABAergic neurons are labeled with YFP-Venus as described previously [27] and present a characteristic distribution throughout all hippocampal layers (Figure 1A,C,E,G).

To directly distinguish between glutamatergic PCs and GABAergic INs, particularly important for cell sorting, we took advantage of a double transgenic VGAT-YFP x NexCre-Ai9 mouse line, generated by crossing VGAT-YFP mice with NexCre-Ai9 mice [28,29]. In this line, GABAergic INs are fluorescently labeled with YFP-Venus and the majority of glutamatergic neurons are labeled with the red fluorescent protein (RFP), TdTomato (Figure 1B,D,F,H).

### 2.2. Cellular Localization of Mature and Precursor BDNF Proteins in the Mouse Hippocampus

To characterize the cellular distribution of mBDNF and proBDNF in both glutamatergic and GABAergic neurons, we performed immunolabeling for these two forms and analyzed the cellular localization pattern of each protein using a confocal microscope.

#### 2.2.1. Cytoplasmic Immunoreactivity of the Mature BDNF Protein in Both Principal Cells and Interneurons

To specifically detect mBDNF, we used antibodies that recognize the 13 kDa protein, which occurs after proBDNF has been processed by the Golgi apparatus. Immunolabeling showed a dispersed cytoplasmic distribution of mBDNF in the cell bodies of glutamatergic neurons in the stratum pyramidale of CA1 and CA3 and also in granule cells of the dentate gyrus (DG) (Figure 2A,B,D,E), as expected [22,30,31]. We also saw strong mBDNF immunoreactivity (IR) in the soma of YFP-positive GABAergic INs distributed across all hippocampal layers (Figure 2A,B). While this strong signal demonstrated the presence of mBDNF in INs, it did not indicate synthesis in these cells, as uptake of mBDNF released by PCs may still be the source of the signal [32,33].

To assess whether INs are post-processing BDNF, we immunolabeled coronal sections for the cis-Golgi specific marker GM130 and looked for colocalization of GM130 and mBDNF. Double labeling for both mBDNF and GM130 revealed at least a partial colocalization of the two proteins within the soma, supporting the idea that BDNF might also be trafficked through the endoplasmic reticulum (ER)–Golgi system in INs (Figure 2C,F–J). As expected, not all mBDNF IR was restricted to somatic compartments, but it was also found distributed throughout neuronal processes, in neuropils of both the stratum radiatum and stratum moleculare (Figure 2A,B). Quantification revealed that 26.6% ± 6.0% of mBDNF colocalized with GM130 (Table 1).

Regardless of its origin, these data demonstrated the presence of mBDNF protein not only in glutamatergic PCs, but also in the vast majority of hippocampal INs. 

#### 2.2.2. Perinuclear Expression of the Precursor BDNF Protein Form in Both Principal Cells and Interneurons

The presence of mBDNF protein in the majority of GABAergic INs in the hippocampus raises the question whether the mature protein can be endogenously synthesized by INs. Therefore, we next investigated the presence of proBDNF in hippocampal neurons by immunofluorescence labeling.

To identify proBDNF, we used an antibody that recognizes the 34 kDa form of the uncleaved precursor protein. Labeling for proBDNF was found in the somata of RFP-labeled glutamatergic neurons and in the cell body layers of all hippocampal areas, as well as YFP-labeled GABAergic INs, distributed throughout the hippocampus (Figure 3A,B,D,E). Images at higher magnification revealed a perinuclear localization of IR in all neurons (Figure 3D–J). Quantification of this colocalization showed that, of the 321 YFP-neurons counted, 287 neurons were positive for proBDNF, corresponding to 89% of the population (data from the dorsal hippocampus of two animals; Table 2). Thus, the vast majority of INs in the hippocampus showed proBDNF expression.

To determine whether the perinuclear localization of proBDNF corresponds to the location of the ER, we performed double immunolabeling for proBDNF and lamin-B1 (Figure 3C,F–J), which labels the inner membrane of the nuclear envelope and perinuclear ER cisternae (Figure 3I). Next, we examined the signal overlap of the two proteins in the dendritic layers of CA1 and CA3, where many of the cell bodies of GABAergic INs are situated. This quantification showed that 56.5% ± 5.9% of the proBDNF signal colocalized with lamin-B1 (Figure 3J; Table 1).

Our data demonstrated that proBDNF is present not only in hippocampal PCs, but also in hippocampal INs and, tentatively, that proBDNF is also likely to be synthesized by INs, given the strong colocalization of lamin and proBDNF in their somata (Figure 3G).

To better understand how proBDNF and mBDNF expression colocalizes with GABAergic neurons, we performed a comparison of proBDNF and mBDNF immunostaining across cell body layers (CA1, CA3 and DG) of the hippocampus (Figure 4). We found that proBDNF was largely perinuclear and localized mainly in somatic regions (Figure 4B,D,F,H), while mBDNF was distributed throughout the whole cell, as expected (Figure 4A,E,G). Despite this difference, both the proteins showed a very strong and consistent localization not only to principal cells, but also to GABAergic neurons, throughout the hippocampus.

### 2.3. Expression of BDNF mRNA in Hippocampal Glutamatergic and GABAergic Neurons Revealed by LCM and RT-qPCR

To confirm the ability of INs to synthesize BDNF, we next examined whether inhibitory INs possess mRNA for BDNF by using LCM combined with real-time RT-qPCR (Figure 5A). We employed a modified implementation of the single-cell LCM approach, previously described by Boone et al. (2013) [34], enabling the isolation of identified fluorescent cell bodies from tissue sections obtained from transgenic mouse lines.

To minimize RNA degradation, we avoided histochemical or immunocytochemical procedures and took advantage of the endogenous YFP and RFP signals found in GABAergic and glutamatergic neurons, respectively (Figure 5B). To ensure reliable dissection of single cells, we prepared 10 µm coronal hippocampal cryosections, thus minimizing the chance of cell stacking. YFP^+^ and RFP^+^ cell bodies were visually identified before LCM. Individual cells were extracted by adhering them to a sterile cap, avoiding the collection of neighboring tissue (Figure 5B, insets). We separately collected cellular/neuronal and neurite/neuropil fractions from different hippocampal layers. For each repetition, the collected samples of fractions included: (1) somata from ~20 GABAergic cells from stratum oriens and stratum radiatum of CA1 and CA3 regions, (2) somata from ~20 PCs from CA1 and CA3 stratum pyramidale, (3) somata from ~20 granule cells from the cell body layer of the DG and (4) cell-body free neuropil tissue from stratum radiatum of CA1 and CA3 in a form and quantity similar to the cell-containing fractions. The last sample was used as a negative control, in which the expression of genes examined in this study was generally unexpected, with the exception of glial genes. The isolation efficiency was high for all samples (Figure 5B). No other tissue was removed from the hippocampal section and the remaining tissue remained intact following the capture procedure (Figure 5B). Collected LCM samples from six transgenic animals were processed immediately for single-cell RNA isolation and reverse transcription (RT) into cDNA. cDNA from the samples was used to detect BDNF mRNA using the qPCR technique. We chose a set of PCR primers specific to exon IX BDNF, which detects BDNF in any of its known isoforms. To distinguish between GABAergic and glutamatergic neurons, we applied primers specific to genes selectively expressed by these two main neuronal groups: (1) Emx1, exclusively expressed in cortical glutamatergic neurons and (2) GAD67, the GABA synthesizing enzyme in INs, which, however, is also expressed at low levels in hippocampal pyramidal and granule cells. Since Emx1 is not expressed by GABAergic neurons [35,36], we were able to estimate the purity of our IN samples. Additionally, glial fibrillary acidic protein (GFAP) primers were used as astrocyte markers. As a reference, we used the neuronal housekeeping gene: class III β-tubulin.

The transcripts for BDNF, Emx1, GAD67 and tubulin genes showed detectable levels with consistent counts from the cell body samples. Counts for these neuronal transcripts were normalized to the geometric mean of the housekeeping gene (Figure 5E). Representative PCR quantification curves for BDNF, Emx1, GAD67 and tubulin transcripts from neuronal cell body and neuropil samples are shown in Figure 5C,D. Next, we calculated the delta crossing point (delta-Cp) for the target genes (BDNF, Emx1 and GAD67) and reference gene (class III β-tubulin; Table 3).

In fractions containing cell bodies of glutamatergic pyramidal and granule cells, we detected comparable amounts of BDNF transcripts with mean delta-Cp values of 0.690 ± 0.393 and 0.585 ± 0.049, respectively. This data confirmed that glutamatergic neurons can synthesize BDNF. The glutamatergic cell body fractions also contained, as expected, the Emx1 gene product, selectively expressed by excitatory cells in amounts of 1.890 ± 0.946 for pyramidal neurons and 0.027 ± 0.004 for granular neurons. The lower amount of Emx1 in dentate granule cells corresponds to the fact that this gene is preferentially and highly expressed by pyramidal neurons [35]. Further, in the glutamatergic samples, we also detected the product for the GAD67 gene at the level of 0.080 ± 0.060 in pyramidal neurons and 1.376 ± 0.169 in granule cells. The high amount of GAD67 observed in dentate granule cells confirmed previous findings that these cells, despite being glutamatergic, express GAD67 and can co-release GABA [37,38,39].

In the fractions containing the cell body of GABAergic INs, BDNF transcripts were also detected with a mean delta-Cp of 0.120 ± 0.073. This finding supported our immunocytochemical data and confirmed the ability of hippocampal INs to synthesize BDNF. BDNF transcript level showed a tendency to be lower in GABAergic samples, compared to the glutamatergic ones, but applied statistic tests did not indicate significant differences between pyramidal cell and IN samples (*p* = 0.2353, Kruskal–Wallis). When neuronal samples were compared with neuropil samples, we found significant differences (*p* = 0.0296 for GABAergic neurons, *p* = 0.001 for pyramidal neurons and *p* = 0.0033 for granular neurons, Kruskal–Wallis; Figure 5E; Table 3). These differences between neuronal and neuropil fractions suggested a preferential somatic accumulation of BDNF transcripts in hippocampal neurons. The highest level of GAD67 transcripts was detected in the GABAergic cell body samples with a mean delta-Cp of 3.350 ± 0.935, as expected. The purity of the GABAergic samples was controlled and confirmed by the lack of Emx1 (Table 3).

In samples of neuropils, we did not detect any product for the BDNF gene (Figure 5C–E; Table 3). In these samples, we also did not detect any transcripts for Emx1 and GAD67, indicating that we were able to collect neuronal cell body-free neuropil samples in a reproducible manner. This data supported the notion that BDNF RNA is primarily expressed at the somatic level. In two neuropil samples, we could detect GFAP transcripts, indicating the presence of astrocytes. There was no difference in the level of BDNF gene product in the samples with and without GFAP signal, supporting proposals that under physiological conditions BDNF is expressed by neurons only [40,41].

In summary, our findings confirmed the presence of BDNF mRNA in hippocampal PCs, but also revealed the presence of its transcript in hippocampal GABAergic INs, indicating that both cortical neuronal classes are able to synthesize BDNF.

### 2.4. Expression of BDNF mRNA in Cortical GABAergic and Glutamatergic Neuron Populations Confirmed by FACS and RT-qPCR

To further assess BDNF expression in GABAergic and glutamatergic neurons, we next used fluorescence cell sorting to purify and isolate fluorescent neurons from the cortex and hippocampus of VGAT-YFP x NexCre-Ai9 mice (12 animals in total) [42]. This resulted in highly pure GABAergic and glutamatergic cell populations (>97%, Figure 6 and Figure 7B). From sorted cells at P0 (6 animals), we prepared monolayer cell cultures of both YFP-labeled GABAergic and RFP-labeled glutamatergic neurons (Figure 6A,G and Figure 7B). The cultures were left to grow and mature for 7–9 days (7–9 DIV) and then processed for either immunocytochemistry or RNA extraction, cDNA preparation and finally, RT-qPCR. A qualitative screening of the cultures with YFP-labeled neurons did not show any contamination with RFP-labeled neurons and vice versa (3–5 coverslips per culture; Figure 6A,G and Figure 7B, top images). To further confirm the purity of isolated neuronal fractions, we performed double immunolabeling for GAD67 (GAD1) and Emx1. The labeling showed no Emx1 in YFP^+^ cultures and no GAD67 in RFP^+^ cultures (Figure 7B, middle images). Finally, we performed immunolabeling for GFAP, an astrocytic marker, but found no signal in any of the neuronal cultures (Figure 7B). This qualitative data confirmed that we obtained two cortical neuron populations with high purity with the FACS approach.

Next, we performed labeling for the mature and precursor BDNF proteins in these cultures. Consistent with the labeling in the perfused material, mBDNF displayed a distributed cytoplasmic labeling (Figure 6C,F,I,L). In contrast, proBDNF immunolabeling showed a tight perinuclear labeling in both YFP-labeled GABAergic (Figure 6B,E) and RFP-labeled glutamatergic neurons (Figure 6H,K). This result further strengthened the conclusions that both mBDNF and proBDNF are present in the two classes of cortical neurons and proBDNF localizes to the rough endoplasmic reticulum, the main locus of protein synthesis.

From a set of cultures (6 animals), we extracted RNA and processed it for RT-qPCR to evaluate the neuronal BDNF, Emx1 and GAD67 transcript levels in dissociated, cortical, YFP-labeled GABAergic and RFP-labeled glutamatergic samples (Figure 7E,F). In samples of glutamatergic neurons, we detected BDNF gene product with a mean delta-Cp of 0.047 ± 0.013 (Table 4). In glutamatergic samples, we also detected Emx1 transcripts with a delta-Cp of 0.007 ± 0.005 and GAD67 transcripts with a delta-Cp of 0.008 ± 0.005. As discussed above, GAD67 was expected in RFP-labeled cortical glutamatergic neurons, because this fraction contains the granule cells, known to express GAD67 (see Section 2.2). In samples of GABAergic neurons, BDNF transcripts were also detected, albeit at a lower level with a delta-Cp of 0.009 ± 0.004. The differences in the amount of BDNF transcripts in GABAeric and glutamatergic cultured neurons were statistically not significant (*p* = 0.1094; two-way ANOVA). GABAergic neurons highly expressed the GAD67 gene, with a mean delta-Cp of 1.239 ± 0.405, substantially higher than the glutamatergic fraction (Table 4). The purity of sorted and cultured GABAergic samples was confirmed by the lack of Emx1 gene expression in all, but a single GABAergic sample at a very low level (delta-Cp = 0.00006; Table 4).

FAC-sorted neurons from P14-P22 mice (two animals at stage P14 and one at stage P22) were processed immediately for RNA extraction, amplification and RT-qPCR of the target (BDNF, Emx1 and GAD67) and reference (tubulin) genes. In all samples of YFP-labeled GABAergic and RFP-labeled glutamatergic fractions, BDNF mRNA was detected at discernible levels (mean delta-Cp = 0.020 ± 0.010 and 0.066 ± 0.024, respectively). The difference in BDNF transcripts was comparable to the results obtained from LCM-captured cells, and the applied statistic test (described in Section 4.6.1) indicated a marginally significant difference (*p* = 0.0406) between the two neocortical neuron populations. In all three samples of RFP-positive glutamatergic cortical neurons, Emx1 and low amounts of GAD67 transcripts were detected (Figure 7C,D,F; Table 4). In contrast, YFP-positive GABAergic samples showed high levels of GAD67 transcripts, but not detectable or very low levels of Emx1 mRNA (Figure 7C–E; Table 4). In addition, GFAP mRNA was not detected in YFP-labeled GABAergic neurons (Figure 7E). We noted a difference in the total mRNA amount from the FACS experiment, compared to the slice LCM experiments performed previously. This was due to the dissociation treatment, which may affect gene expression. Even if the absolute delta-Cp values of BDNF, Emx1 and GAD67 were different, the relative amounts of the investigated transcripts remained comparable between the two data sets obtained with LCM and FACS.

In summary, these results further confirmed that cortical GABAergic INs consistently express the BDNF transcripts, albeit at lower levels compared to glutamatergic principal neurons.

## 3. Discussion

In this study, we provide compelling evidence for BDNF gene expression not only in PCs, but also in GABAergic INs of the hippocampus and the neocortex. Our immunocytochemical findings demonstrated that, in addition to the mature protein form of BDNF being present in both GABAergic and glutamatergic neurons, its precursor form, proBDNF, was also present in the majority of INs and PCs examined. Importantly, proBDNF, with its perinuclear distribution, colocalized strongly with lamin, which selectively labels the nuclear envelope and perinuclear ER cisternae, the site of protein synthesis in the ER. Finally, using independent cell purification techniques, LCM in the tissue slices and FACS of dissociated cells, to specifically purify hippocampal and cortical PCs and INs, we could robustly demonstrate the presence of BDNF transcripts in both neuronal populations and thus, finally confirm the presence of BDNF mRNA in GABAergic neurons.

### 3.1. Technical Considerations

LCM is a technique increasingly employed for mRNA expression studies in physiological and pathological conditions and it allows for targeted sampling of specific neuronal subpopulations from brain tissue samples [34]. In our study, we adapted this method to obtain samples of glutamatergic PCs and GABAergic INs from the hippocampus for BDNF gene expression profiling by combining it with RT-qPCR. During the procedure, we aimed to minimize RNA degradation, by omitting histochemical or immunocytochemical processing of the tissues, to visualize specific neuron types; instead, we relied on endogenous fluorescence marker expression in transgenic mouse lines. Additionally, we shortened the time to dissecting the tissue slices, significantly reducing the total number of recovered cells per experiment. The modified LCM approach can, thus, be used to efficiently collect individual cell populations or even single cells from very thin, methanol-fixed sections from transgenic mice expressing fluorescent markers.

We used LCM to focus on BDNF expression in the cell bodies of cortical neurons. To control for potential mRNA contamination from neuropils, we took samples of the surrounding tissue making sure that no neuronal cell bodies were captured. Indeed, in these samples, we did not detect Emx1 and GAD67 transcripts. We detected the expression of GFAP gene product in some of them, plausibly due to the presence of astrocytes and their processes, which occupy approximately 5% of the neuropils in the CA1 stratum radiatum [43,44], intermingling with dendrites, axons and synapses; however, independent of the presence or absence of GFAP signal, BDNF was not detected in these control samples from neuropils.

Real-time RT-qPCR is a well-established method for the analysis of DNA and RNA molecules, even in previously fixed samples [45]. For the detection of gene amplification, we used SYBR Green in the real-time PCR assay, which provides a rapid and accurate alternative to the ‘TaqMan’ approach [46,47]. SYBR Green is a non-coupled fluorescent dye and generally binds to any dsDNA [48]. All primers, for the gene of interest, used in our study were empirically validated by doing an actual RT-qPCR experiment and inspecting the melting point curve. In addition, a standard curve for individual primers was run in order to estimate the efficiency of the PCR primers. All primer pairs were designed carefully such that they could not non-specifically amplify other genomic targets. For BDNF, our gene of interest, we chose a primer optimized for the amplification of exon IX BDNF, since it is known that the BDNF gene comprises nine exons, but the coding sequence (CDS) resides solely in exon IX, with the other exons being involved in protein shuttling and subcellular localization [10]. Thus, the eight upstream exons drive the transcription of multiple BDNF splice variants that encode an identical BDNF protein in a regional and cell type-specific manner [8,49]. Since, we wanted to analyze the somatic gene expression in two main neuron populations, we chose the somatic markers, Emx1 and GAD67, for glutamatergic and GABAergic neuron types, respectively. As the reference gene, we used class III β-tubulin, which is known to be expressed in all neuron types [50].

### 3.2. BDNF Precursor Protein Localized not only to Glutamatergic Principal Cells, but also to GABAergic Hippocampal Interneurons

In the present study, we showed that proBDNF protein, a precursor of the mature form of BDNF, is present in the two main hippocampal neuron populations, glutamatergic neurons and GABAergic INs. Moreover, the immunolabeling for proBDNF highly overlapped with that of lamin, a marker of the nuclear envelope and ER perinuclear cisternae, the area where protein synthesis takes place. We take these results as a strong indicator that proBDNF could be synthesized in both of these neuronal groups.

The mature form of BDNF has been previously detected in cortical GABAergic INs [51]. Our present results confirmed this finding and demonstrated that the majority of hippocampal INs show substantial levels of immunoreactivity for the ~14 kDa mature protein. It has been shown that BDNF can be taken up by neurons from the extracellular space [14,52]. As principal cells produce and release high levels of BDNF [2,22], the ability of cortical INs to synthesize the protein has remained an open question [23,24].

Our immunocytochemical results now provide further evidence for the possible synthesis of BDNF in INs. First, the 34 kDa proBDNF protein was also found to be present at high levels in ~84% of hippocampal INs. The proBDNF protein is generally regarded as an intracellular precursor of BDNF, which can be cleaved by furin (the abundant protein convertase present in all cells) or pro-convertases to produce the secretable mature form [5,53]. There is some evidence that proBDNF might also be released and an alternative extracellular cleavage by metalloproteinases and plasmin may exist [53,54,55]. Thus, endocytosis, similar to that of mBDNF, may explain the observed presence of proBDNF in INs, if readily released by hippocampal PCs [56]. A contradicting observation, however, argues that dentate gyrus GABAergic INs are unable to respond to proBDNF [57,58], thus may not be able to take up this form of the protein, even if available in the extracellular space. This is consistent with the fact that proBDNF can interact only with the p75 receptor [57], but most INs, except for the parvalbumin ones, are devoid of the receptor [57,59]. In fact, this receptor would be needed to mediate the endocytotic uptake of proBDNF, too.

The second piece of our immunocytochemical results which points to a possible synthesis of BDNF was the perinuclear colocalization of proBDNF with lamin, in the area where protein synthesis takes place. This was in stark contrast to the more diffuse cytoplasmic localization of mBDNF observed in our study. Indeed, if proBDNF was taken up by INs from the extracellular space, a similar cytoplasmic or an endosomal localization would be expected. While not a definitive evidence, our data on the perinuclear localization and the high level of proBDNF, comparable to that detected in PCs, convergently suggested the endogenous production of the protein in hippocampal INs.

### 3.3. BDNF RNA is Expressed in Both Glutamatergic Neurons and GABAergic Interneurons in the Cortical Structures.

It is generally assumed that only glutamatergic PCs express BDNF mRNA [2,17,22,60]. In our study, we showed that hippocampal GABAergic INs express BDNF mRNA and, thus, can produce BDNF themselves, further supporting our immunocytochemical evidence. This finding also converged with results of a recent in situ hybridization (ISH) study demonstrating that, while endogenous BDNF transcripts primarily localized to the somatic compartments of putative principal neurons in pyramidal and granular cell layers, they can be also detected in neurons scattered in the dendritic layers of hippocampal subfields [23]. While the authors did not identify the latter as GABAergic neurons, the high number of these scattered labeled cells indicated that they must include at least some types of INs [61,62]. Indeed, in our VGAT-YFP and VGAT-YFP x NexCre-Ai9 transgenic mice, we observed very little RFP-positive, displaced, glutamatergic neurons in the dendritic layers, the stratum oriens and radiatum, but high numbers of YFP-positive GABAergic INs were observed.

We also used LCM to dissect the somata of YFP-positive INs from these two dendritic layers to avoid contamination by glutamatergic PCs and compared these to YFP-negative and RFP-positive samples of PCs from the cell body layers of the CA1 and CA3 areas, and the DG. An RT-qPCR assessment of these samples confirmed the expression of BDNF transcripts, as well as the glutamatergic marker Emx1 in PCs, whereas GAD67 expression was absent or very low in CA1 and CA3 PCs. In dentate granule cells, GAD67 expression was intermediate high, consistent with the fact that these cells can, in addition to glutamate, co-release GABA [37,38,39]. In contrast, in our sample of INs, we reliably obtained signals for BDNF RNA and for the GABA-synthesizing enzyme GAD67, whereas Emx1 was absent from most of these samples indicating the lack of contamination by glutamatergic PCs. Finally, our RT-qPCR results of control samples obtained from cell body-free hippocampal neuropils showed no traces of BDNF, Emx1 or GAD67, providing support to the validity of our samples. Our results obtained from FAC sorting showed converging results and supported the notion that cortical INs can express BDNF mRNA transcripts.

Our data further showed that endogenous expression of BDNF was relatively low, especially in INs, compared to the neuronal marker and primarily localized to the somatic compartment. Thus, our results did not support the idea of substantial BDNF mRNA transport to dendritic and axonal processes [33,63,64]. The absent signal in neuropils implied that, under basal (unstimulated) conditions, there was a limited potential for local translation of BDNF mRNA. The lack of BDNF signal in our samples from neuropils further suggested that BDNF mRNA is not present in astrocytes. Even in samples of neuropils in which GFAP mRNA was detected, indicating the presence of harvested astrocytes, BDNF gene product was consistently absent. These findings were in line with published data that muscimol manipulation cannot upregulate BDNF mRNA in glial cultures [40].

Our data from LCM and FACS supported the immunocytochemistry results and they convergently suggested that the two main cortical neuron classes, glutamatergic and GABAergic, can synthesize BDNF.

### 3.4. Functional Implications for BDNF Synthetized By Interneurons

The evidence that cortical GABAergic INs can also produce BDNF, presented in this study, raises new questions about possible autocrine and paracrine actions of mature BDNF and its precursor. This question fits into the broader debate on whether proBDNF can be secreted and has any biologically relevant activity. In fact, previously, it was assumed that only the mature form of BDNF could be secreted and was biologically active, whereas proBDNF, localized intracellularly, served only as an inactive precursor. Accumulating evidence shows that both proBDNF and mBDNF can be released and are active, with important functions in various developmental and physiological processes [65,66,67]. Moreover, the proBDNF and mBDNF forms have been suggested to serve opposing effects via the p75 neurotrophin receptor (p75NTR) and TrkB receptors, respectively [65,66,67]; while mature BDNF promotes neuronal survival, differentiation, synaptic plasticity and long-term potentiation (LTP), proBDNF may induce apoptosis and growth cone retraction, reduce dendritic spine density and facilitate long-term depression (LTD) in hippocampal slices [68]. The proBDNF form has been further shown to reduce the intrinsic excitability of pyramidal cells of the entorhinal cortex and can thereby modulate memory functions and seizure propensity [69].

The effects of BDNF on the GABAergic system appear to be more diverse. Mature BDNF can enhance, but also reduce, inhibitory synaptic transmission by modulating presynaptic GABA release [70] or by up- or down-regulating postsynaptic GABA_A_ receptor expression [71,72]. A recent study also found a bidirectional effect of proBDNF on GABAergic synaptic activity in the hippocampus, which was dependent on NMDA receptor activation [73]. Beyond synaptic transmission, mature BDNF has also been found to decrease the excitability of GABAergic INs via activation of TrkB, whereas no cellular effects were observed for proBDNF [57]. While these diverse effects may reflect developmental stage- and brain region-specific actions of the neurotrophins [74], an important aspect to consider is the morphological, physiological and molecular heterogeneity of the GABAergic system. In fact, the large number of IN types [61,62] would require a rigorous, systematic analysis of the modulatory effects of both BDNF and the precursor protein on their cellular and synaptic functions.

In summary, in the present study, we provided evidence indicating that BDNF can be synthesized as a precursor in the two main neuronal populations of the mouse hippocampus and neocortex. While proBDNF is likely to constitute an intermediate step in the synthesis of BDNF, it may also provide a neurotrophic environment necessary for the continuous degenerative/regenerative process inherent in the hippocampal GABAergic system; however, the mechanism by which endogenous proBDNF can affect INs under competitive environments needs to be explored.

## 4. Materials and Methods

### 4.1. Animals

For morphological analysis and LCM and FACS experiments, we utilized a total of 18 (6 for LCM and 12 for FACS) early postnatal and young adult male transgenic mice (~2 to 5 months old, 25–30 g for immunocytochemistry and LCM; P0-P22 for FACS) from two lines: (1) VGAT-YFP [27] and (2) VGAT-YFP x NexCre-Ai9 mice [28,70]. The breeding method was described by Turko et al. (2019) [42]. In order to identify cortical GABAergic INs, we used VGAT-YFP-positive mice expressing an improved yellow fluorescent protein (YFP-Venus) under the promoter of the vesicular GABA transporter (VGAT). To differentiate between INs and principal glutamatergic neurons, we employed double-positive VGAT-YFP × NexCre-Ai9 co-expressing both Venus in GA/BAergic neurons and TdTomato (RFP, red fluorescent protein) in postmitotic glutamatergic neurons under the control of the CAG promoter, making it possible to fluorescently identify and to sort these specific cell types of interest. Both transgenic lines were obtained from local animal breeding of the Charité and exhibited normal growth and reproductive behavior.

All procedures, including animal handling and maintenance, were performed in accordance with the regulations of the animal welfare committee of Charité Berlin (Germany), the National Act on the Use of Experimental Animals (Germany), local authorities (LaGeSo Berlin, registration number: O-0098/12) and the European Council Directive 86/609/EEC (approved on 14 05 2012).

### 4.2. Tissue Preparation for Microscopic Analysis

For cellular and subcellular protein distribution analysis, we used well-preserved tissues after perfusion-fixation. The animals were anesthetized with a mixture of 50 mg/kg ketamine (Actavis) and 20 mg/mL xylazine (Rompun; Bayer Health Care, Berlin, Germany) and perfused transcardially with a fixative containing 4% paraformaldehyde (PFA; Electron Microscopy Sciences, Hatfield, PA) with 0.2% picric acid (Fluka Chemie, Buchs, Switzerland) in a phosphate-buffered solution (0.1 M PB, pH = 7.2). The brains were removed and dissected into blocks containing the hippocampus using a coronal rodent brain matrix (ASI Instruments, Warren, USA) and were processed for light microscopy as previously described by Münster-Wandowski et al. (2017) [75]. In all morphological experiments, we analyzed 3–5 coronal sections of the dorsal hippocampus from each animal.

### 4.3. Immunofluorescence Labeling

In order to analyze the distribution of precursor and mature BDNF proteins in neocortical structures, we applied immunocytochemical labeling combined with confocal analysis. For a comprehensive list of the antibodies, including their characteristics, dilution and source, please refer to Table 5. We chose the antibodies against precursor and mature BDNF proteins based on their target’s respective molecular weight. The endogenously synthesized proBDNF has a molecular mass of ~32 to 34 kDa, in contrast, mature, biologically active BDNF has a molecular weight of ~13 to 14 kDa. According to the Bioss datasheet (Table 5), the antibody used to detect mature BDNF protein specifically recognized one band in the ~13 to 14 kDa range, corresponding to the weight of mBDNF monomer. The antibody against the precursor form protein was first used in a single immunofluorescence analysis to determine the distribution profile of this protein in mouse hippocampus and hippocampal cell culture (see Section 4.7). For the double immunostaining analysis for proBDNF and mBDNF, cleaved form was used to determine the cellular and compartmental localization of these molecules in both mouse hippocampal sections and hippocampal cell culture. To confirm the specific subcellular localization of the precursor and mature form, we applied double labeling for proBDNF and lamin-B1 of the inner nuclear membrane and for mBDNF and the Golgi marker, GM130.

Double and triple immunostaining methods for proBDNF and several specific GABA, glutamatergic and astrocytic cellular and presynaptic markers were used to confirm the purity of sorted GABAergic and glutamatergic cell cultures (Table 3; Section 4.7).

#### Control Experiments

We carefully validated BDNF isoform immunolabeling by focusing on proper negative and positive controls. Negative staining controls for all immunofluorescence procedures were performed by substitution of non-immune serum for the primary or secondary antibodies. As a positive control, we examined the immunofluorescence of BDNF in glutamatergic cells in both hippocampal tissue and cell culture.

### 4.4. Confocal Imaging

Coronal hippocampal sections and sorted cells growing on coverslips were imaged on a laser scanning confocal microscope (FV1000, Olympus, Japan). In order to get an overview of proBDNF and mBDNF distribution in the mouse hippocampus, we imaged 20 µm thick coronal hippocampal sections at low magnification (4x air immersion; Olympus, Japan) and arranged the overview images. Higher resolution image stacks were acquired using an x60 silicon oil immersion lens (0.5 µm step size, a numerical aperture of 1.30; UPlanSApo, Olympus) with a zoom factor of either 1 or 4 to resolve precise subcellular localization. Excitation wavelengths were 488 nm for anti-mouse Alexa Fluor-488 (Invitrogen, Karlsruhe, Germany), 405 nm for anti-guinea pig Alexa Fluor-405 (Jackson Immuno Research, West Grove, PA, USA), 643 nm for anti-rabbit Alexa Fluor-647 (Life Technologies, Darmstadt, Germany), 515 nm for the yellow fluorescent protein (YFP) [76] and 555 nm for the red fluorescence protein (RFP). The images were acquired through separate channels and temporally non-overlapping excitations of the fluorochromes, and analyzed off-line using the ImageJ software package (courtesy of W.S. Rasband, U.S. National Institutes of Health, Bethesda, Maryland, http://rsb.info.nih.gov/ij/). The analysis of colocalization was performed in the Fiji/ImageJ software (an open source image processing package) using a pixel method based on the isodata algorithm [77].

### 4.5. Laser Capture Microdissection, RNA isolation and Reverse Transcription

In order to obtain samples of identified neuronal populations, we utilized the laser capture microdissection method using an inverted epifluoreescence microscope (Arcturus XT Laser Capture Microdissection Instrument, Termo Fisher Scientific, Germany) equipped with infrared and UV lasers and controlled by its own (Arcturus XT) software. The intensity, aperture and cutting velocity were adjusted for both the lasers (infrared and UV). To dissect and sample neurons, the IR laser beam was first pulsed onto the cell bodies of visually selected neurons in the thin sections, causing them to stick to an Arcturus LCM cap placed directly above the sample. This was then followed by careful dissection by pulsing a UV laser beam carefully along the perimeter of the cell bodies (Figure 5B, top inset). GABAergic inhibitory cells were identified by their strong YFP fluorescence and sampled in the stratum oriens of the CA1 and CA3 regions. PCs were collected from the cell body layers of the CA1 and CA3 regions as well as from the dentate gyrus. INs were identified based on Venus-YFP fluorescence, while principal and granular cells were identified based on their location and expression of the TdTomato-RFP protein. Finally, cell body-free control samples were obtained from the neuropils in the CA1-CA3 stratum oriens in the dorsal hippocampus [78,79]. From each of these populations, we collected similar amounts of tissues, approximately proportional to 20–30 dissected cells. Cells and neuropil tissue portions were picked up in serial sections from each field using a 20x magnification objective. The LCM caps with the adhered dissected tissue samples were then removed from the dissection stage by a robotic arm, before 10 µL of an extraction buffer for cell lysis was added. For RNA isolation from small amounts of cells (~20 captured cells per neuron type), we used the commercially available Arcturus® PicoPure® RNA Isolation Kit, which is designed to recover high-quality total RNA from fewer than 10 cells (Applied biosystems by Thermo Fisher Scientific, Vilnius, Lithuania). The caps were placed in microcentrifuge tube caps and RNA was isolated according to the manufacturer’s recommendations, including on-column DNase digestion and elution step. RNA was finally reverse transcribed using the SuperScript™ III Reverse Transcriptase Kit (Invitrogen) according to the manufacturer’s instructions.

Due to the low initial amount of RNA collected (<5 ng) from ~20 cells per group, the total amount of RNA per cell was not estimated. Nonetheless, in order to get comparable results, we captured the same amount of cells for each neuron group and were able to get reproducible and comparable results for housekeeping genes for each neuron type.

### 4.6. Quantitative Real-time PCR

To show the expression of BDNF mRNA in GABAergic INs, we applied quantitative real-time PCR (qPCR). It was performed using the Light Cycler 480 Real-Time PCR Instrument (Roche, Rotkreuz, Switzerland) according to the manufacturer’s instructions. The PCR reactions were performed in a final volume of 20 µL with the LC-DNA Master SYBR Green II mix (Fluocycle, Euroclone). Each primer and RT mixture for all genes were added as the PCR template and water was used as the negative control. The real-time PCR reaction was quantified by recording the number of amplification cycles at which the signal was in the logarithmic phase and reached a threshold above the basal background. The amount of target genes was normalized relative to neuronal class III β-tubulin. Primer sequences specific to the examined genes and the amplification conditions are shown in Table 6. Tubulin was the housekeeping gene used as the reference control during the real-time procedure. The other primers were BDNF exon IX (encodes the protein), GAD67 (glutamate decarboxylase 67, an inhibitory somatic marker), Emx1 (a glutamatergic somatic marker) and GFAP (an astrocytic marker). A relative quantification of target cDNA was performed by using standard curves of serial control cDNA dilutions to calculate the PCR reaction efficiency (Table 6).

When evaluating the number of cycles required to detect mRNA expression in the LCM-RNA samples, we stayed within the quantitative range reaching the steady state in less than 55 cycles reliably, despite the low initial RNA amount. Positive reactions containing RT were performed in duplicates and a negative control lacking the RT enzyme was also run. Without the RT enzyme present, no product was detected by SYBR Green quantitative RT-qPCR (Figure 5C,D), indicating no residual genomic DNA contamination. Additionally, samples containing water as a template were also run and showed no product.

We employed a relative quantification approach by using a reference gene that was inserted in the equation for normalization. Depending on the initial amount of target DNA in the reaction mixture, after a certain number of cycles, the fluorescence signals of the PCR product would stand out statistically significantly from those of the background. This threshold was determined by the software, and the so-called crossing point (Cp) was defined as the number of cycles at which the fluorescence signal exceeded the threshold. We did not apply absolute quantification because we primarily investigated BDNF expression in GABAergic neurons, as proof of principle, and we compared relative BDNF expression data with the glutamatergic and granular neuron population, which is known to express this gene.

#### 4.6.1. Statistical Analysis

Each set of qPCR experiments was performed with 1–2 brains (slices) and repetitions N = 6 (6 individual transgenic animals) for laser-captured cells, N = 3 (each of 2 animals, in total 6 individual animals) for sorted cells at P0 and cultured for 7–9 DIV and N = 3 (each of 2 animals, in total 6 individual animals) for sorted cells at P14-P22. The samples were immediately processed for RNA isolation. Real-time PCR data was quantified using the threshold value calibrated on the reference gene (class III β-tubulin) in the same reaction run. A statistical analysis was performed using GraphPad Prism software version 8.0 (GraphPad Software Inc). To determine the distribution of the samples in each population of captured cells, the Kruskal–Wallis (post-hoc after Dunn) multiple comparison test was used. To analyze the distribution of the samples in two sorted neuron populations, we applied the two-way ANOVA (Holm/Sidak) multiple comparison test. All our tests were conducted using an α-value of 0.05, and differences were considered statistically significant, if the *P*-value was smaller than the α-value (* < 0.05, ** < 0.01, *** < 0.005).

### 4.7. Purification and Culture of GABAergic and Glutamatergic Neurons Obtained by Cell Sorting

The purification and culture of GABAergic and glutamatergic neurons have been described previously (including a Jove video protocol of the whole procedure, found in Turko et al. (2019)). In brief, cortico-hippocampal tissues from NexCre-Ai9x VGAT-Venus mice (5–10 pooled litter mates) were dissected and papain-dissociated (1.5 mg/mL for 30 min) in preparation for cell sorting. Neurons taken for direct RNA extraction and PCR were sorted from animals at postnatal days 14–22. Neurons harvested for cell culture were sorted at postnatal days 0–2. Cells were sorted using BD FACSAria I or II flow cytometers at the Flow Cytometry and Cell Sorting Facility (FCCF) in the Deutsches Rheuma-Forschungszentrum (DRFZ). Sort rates of 900 events/second for GABAergic neurons and 1300–1800 events/second for glutamatergic neurons could be routinely achieved. An example dot plot showing typical gating parameters is given in Figure 6A. Cells were transported and collected in Hibernate A (BrainBits Ltd.) transport medium—supplemented with B27 (1x concentration), Glutamax (1x concentration) and penicillin-streptomycin (100 U/mL), all sourced from Gibco). Cultured neurons were grown in Neural Basal A (NBA) medium, at 37 °C/5% CO_2_ in a humidified incubator (Thermo Fisher Scientific; with the same supplementation as Hiberate A). Cultured cells taken for PCR were grown at a density of 50,000 cells per well on the base of a 24-well cell culture plate. Cells used for immunocytochemistry were plated on round 12 mm glass coverslips, in a 10 µL droplet, at a density of 10,000 cells per coverslip (Glas – Menzel). After 1 h incubation, cells were fed with 500 µL of prewarmed, supplemented NBA medium (37 °C). Coverslips and culture plates were coated in a poly-L-lysine hydrobromide solution (Merck; 20 µg/mL, 1 h). Cells were grown for 7–9 days in vitro (DIV) before direct fixation in 4% PFA solution for immunocytochemistry or treated with an extraction buffer for PCR experiments. We adjusted the concentration of cDNA to minimize residual contamination signals by diluting the samples.

## 5. Conclusions

In our present study, we showed that cortical and hippocampal GABAergic INs contain the mRNA required for the synthesis of BDNF. We also showed the presence of its precursor protein proBDNF in INs with expression levels comparable to those in PCs. This fact raises several questions: Can BDNF be synthesized by all GABAergic IN types? Can proBDNF released from INs or does it serve only as a precursor protein? Which BDNF splice variants are preferentially expressed by INs? Do BDNF splice variants show region-specific expression in INs? Future studies should address these questions.

The changes in BDNF expression under pathological conditions are also not fully understood. Several reports suggest that a reduction in BDNF expression levels is a major factor underlying the emergence of diverse neurological and neurodegenerative disorders, such as Alzheimer’s disease. Indeed, the expression of the precursor protein and mature mBDNF is decreased in the parietal cortex and hippocampus in patients with Alzheimer’s disease [80,81,82,83]. In the light of our results, it would be interesting to explore whether this decrease affects only the glutamatergic population, or also other neuron types, including GABAergic INs.

Interestingly, alterations in the ratio of proBDNF/mBDNF have been described in the brain of individuals with autism, suggesting that the balance between these isoforms could be relevant for neurological and psychiatric disorders [84]. Convergently, a reduction in proBDNF processing and mBDNF secretion correlates with deficits in memory and cognitive decline in mild cognitive impairment [83,85]. Thus, the proBDNF/mBDNF ratio in excitatory and inhibitory neurons may serve as a biomarker for neuropsychiatric and neuropathological conditions.

## Figures and Tables

**Figure 1 ijms-21-01567-f001:**
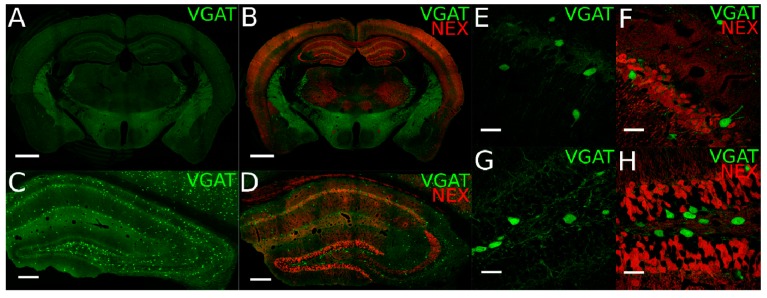
Identification of hippocampal glutamatergic principal cells (PCs) and GABAergic interneurons (INs) in transgenic mouse lines. (**A,B**) Overview images of the yellow fluorescent protein (YFP; green pseudocolor) and red fluorescent protein (RFP; red) reporter signals expressed under the VGAT and NEX (neuronal helix-loop-helix protein) promoters, respectively, in coronal brain slices of (A) single VGAT-YFP and (B) VGAT-YFP x NexCre-Ai9 transgenic mouse lines. (**C**,**D**) Zoomed-in images of the dorsal hippocampus showing either (C) endogenous YFP fluorescent signal for GABAergic neurons or (D) the YFP and RFP fluorescent signal for both GABAergic and glutamatergic neurons. (**E**–**H**) High-power confocal images of hippocampal CA1 area (E,F) and the dentate gyrus (DG; G,H) illustrating the distribution of GABAergic and glutamatergic neurons in the two transgenic lines. Scale bars: (A,B), 1000 µm; (C,D), 200 µm; (E–H), 30 µm.

**Figure 2 ijms-21-01567-f002:**
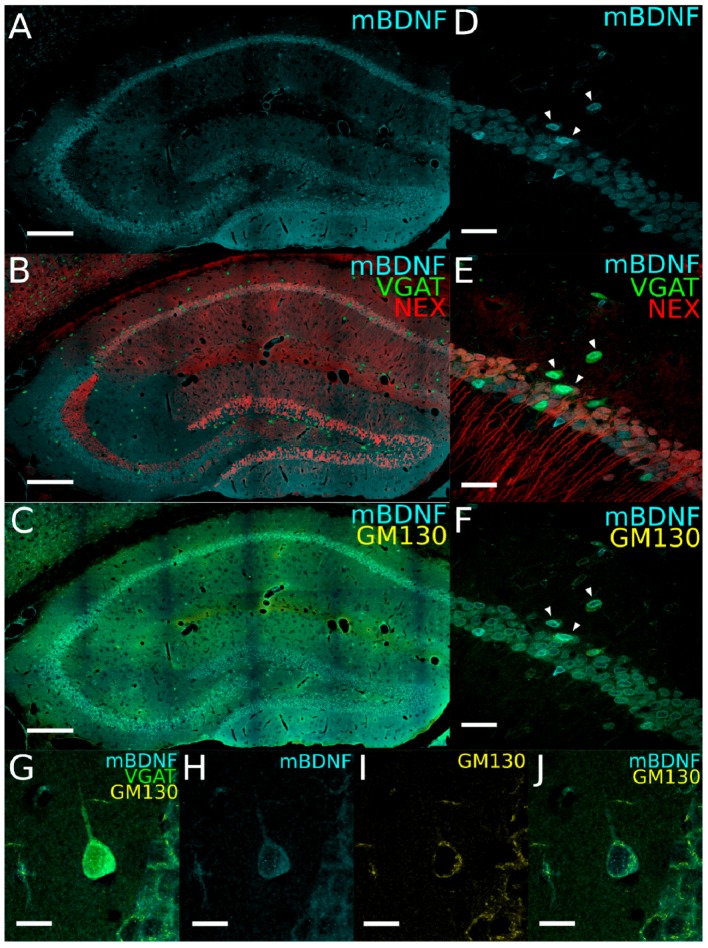
Expression of the mature form of brain-derived neurotrophic factor (mBDNF) in hippocampal glutamatergic and GABAergic neurons. (**A**–**C**) Immunostaining for mBDNF (cyan pseudocolor; A,B) and the Golgi marker, GM130 (yellow; C) in the dorsal hippocampus of VGAT-YFP x NexCre-Ai9 transgenic mice. (**D**–**F**) High-power confocal images from hippocampal CA1 area immunolabeled for mBDNF (D,E) and GM130 (F). Arrowheads indicate a cluster of YFP-positive GABAergic interneurons (INs; green), which are also labeled for mBDNF (D–F). (**G**–**J**) High-power confocal images from a hippocampal YFP-positive IN immunolabeled for mBDNF (G,H) and GM130 (I,J) (cyan and yellow, respectively) in the CA1 area from VGAT-YFP x NexCre-Ai9 transgenic mice (YFP signal in green). Note that the mBDNF signal is distributed throughout the somatic cytoplasm (H) and shows a partial colocalization with GM130 (J). Scale bars: (A–C), 200 µm; (D–F), 30 µm; (G–J), 10 µm.

**Figure 3 ijms-21-01567-f003:**
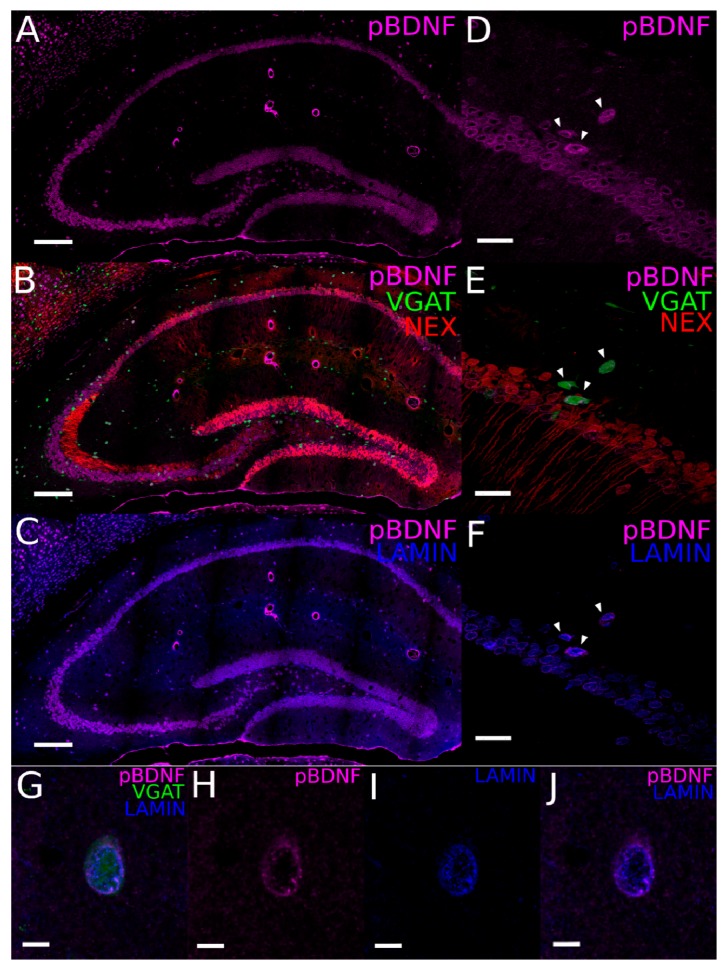
Expression of proBDNF in hippocampal glutamatergic and GABAergic neurons. (**A**–**C**) Immunostaining for proBDNF (magenta pseudocolor; A,B) and the perinuclear marker, lamin (blue; C) in the hippocampus of VGAT-YFP x NexCre-Ai9 transgenic mice. (**D**–**F**) High-power confocal images from the hippocampal CA1 area immunolabeled for proBDNF (D,E) and lamin (F). Arrowheads indicate a cluster of YFP-positive GABAergic INs (in green), which are also labeled for proBDNF (D–F). Note that this section is adjacent to the one shown in Figure 2D–F. (**G**–**J**) High-power confocal images from a hippocampal GABAergic neuron immunolabeled for proBDNF (G,H) and lamin (I,J) (magenta and blue, respectively). Note the perinuclear localization of proBDNF and its colocalization with lamin in (J). Scale bars: (A–C), 200 µm; (D–F), 30 µm; (G–J), 10 µm.

**Figure 4 ijms-21-01567-f004:**
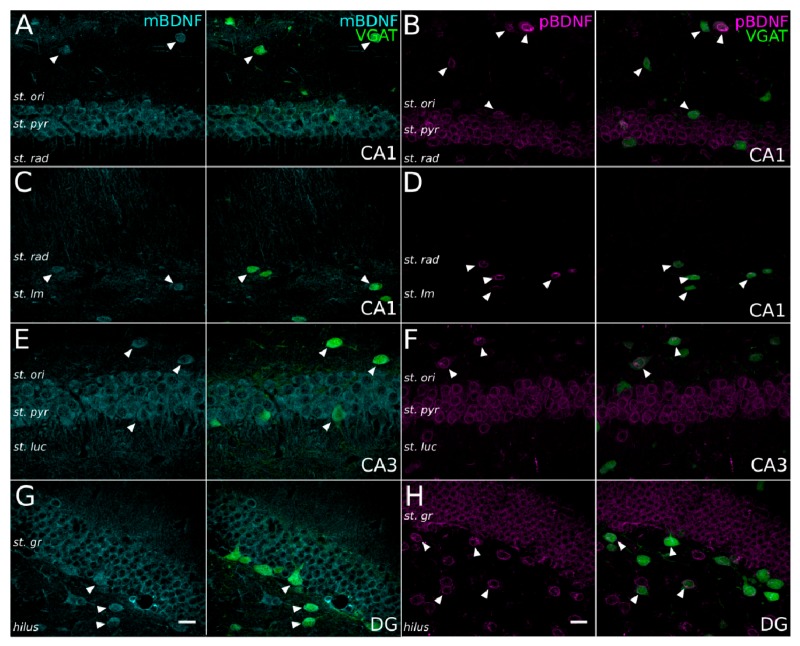
Expression of mBDNF and proBDNF in GABAergic cells in the main hippocampal areas. (**A–H**) Confocal images of immunostaining for mBDNF (cyan pseudocolor; A,C,E,G) and proBDNF (magenta; B,D,F,H) in the CA1 (A–D), CA3 (E,F) and dentate gyrus (G,H) of the hippocampal formation of VGAT-YFP x NexCre-Ai9 transgenic mice (images on the left and right combine the immunostaining and the YFP signal). Arrowheads indicate GABAergic INs labeled for mBDNF or proBDNF. Note that while the mBDNF signal is distributed in the cytoplasm (G), proBDNF signal is mainly localized to the perinuclear region (H). Scale bars: 20 µm.

**Figure 5 ijms-21-01567-f005:**
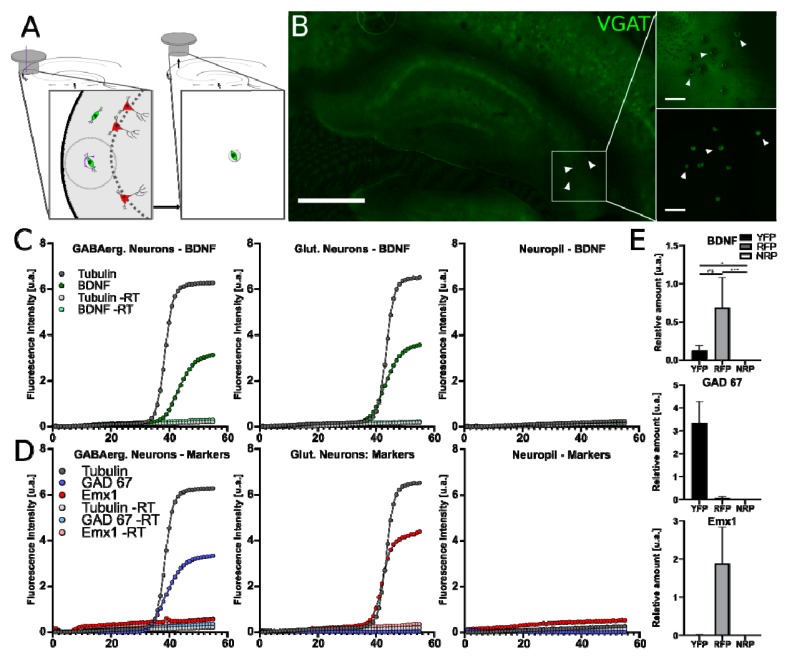
Laser capture microdissection (LCM) and reverse transcription quantitative PCR (RT-qPCR) from microdissected hippocampal glutamatergic and GABAergic neurons. BDNF mRNA quantification from tissue slices following LCM and RT-qPCR: (**A**) Scheme of the LCM procedure. Cells expressing specific reporter genes can be identified through fluorescence microscopy and dissected, and their cell bodies can be harvested from the surrounding tissue. (**B**) Fluorescence image of a tissue section processed for LCM. Insets show the site of the removed sample (top) and the sample (bottom) following the microdissection of GABAergic cell bodies from the stratum oriens (arrowheads). (**C**) Fluorescence intensity vs. cycle plots for RT-qPCR amplification of BDNF and class III β-tubulin, in laser-captured GABAergic neurons (left), glutamatergic neurons (middle) or neuropils (right). (**D**) Intensity vs. cycle plots for RT-qPCR amplification of neuron type-specific transcripts: GAD67 (GABAergic) and Emx1 (glutamatergic), as well as class III β-tubulin (as reference), for the three samples. (**E**) Summary bar charts of the quantification of BDNF, GAD67 and Emx1 target genes normalized to class III β-tubulin, as a neuron-specific “housekeeping” control. Scale bar: (B), 1000 µm; insets, 100 µm.

**Figure 6 ijms-21-01567-f006:**
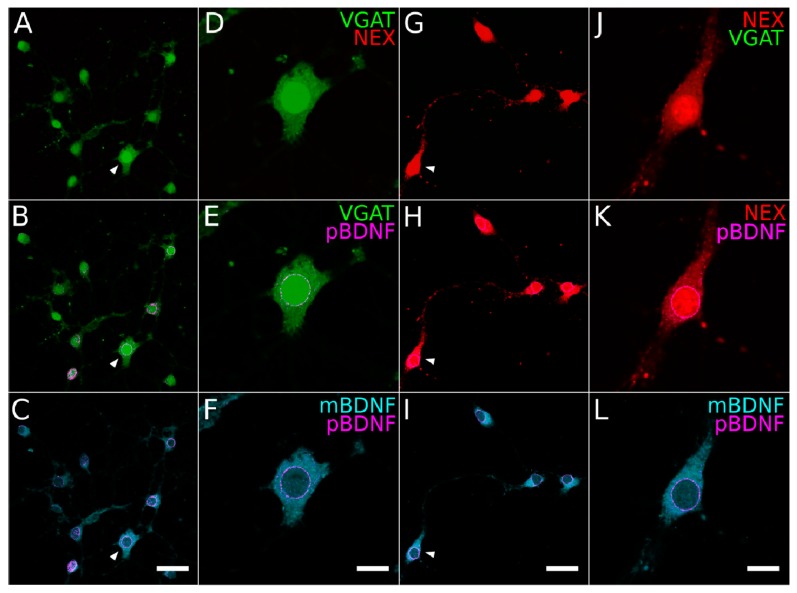
Expression of mBDNF and proBDNF in purified cortical neuron cultures. Immunostaining in (**A–F**) purified GABAergic ((A–C)**,** x30 magnification; (D–F), x60 magnification) and (**G**–**L**) glutamatergic ((G–I), x30 magnification; (J–L), x60 magnification) primary cortical neuron cultures. The fluorescent signal of VGAT (green pseudocolor) and NEX (red) in both types of cultures (A,D,G,J) and overlay with proBDNF (B,E,H,K; magenta) as well as an overlay of mBDNF (cyan) and proBDNF in both populations (C,F,I,L). The arrowheads indicate the cells shown in the x60 magnifications. Note the cytoplasmic distribution of mBDNF compared to the overwhelmingly perinuclear localization of the proBDNF signal (F,L). Scale bar: (C,I), 40 µm; (F,L), 10 µm.

**Figure 7 ijms-21-01567-f007:**
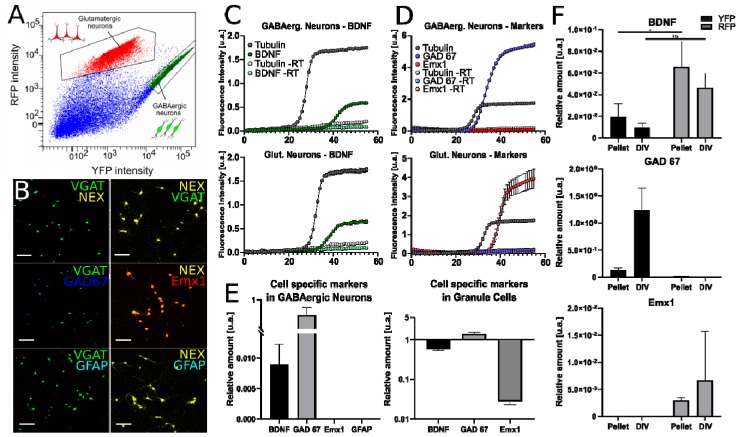
Fluorescence-activated cell sorting (FACS), immunocytochemistry and RT-qPCR from sorted cortical glutamatergic and GABAergic neuron populations. BDNF mRNA quantification from sorted and cultured cell populations: (**A**) Fluorescent scatter of whole neocortex cell homogenate from VGAT-YFP x NexCre-Ai9 transgenic mice with example gates used to segregate both cell populations. (**B)** Immunolabeling of the sorted pure cell cultures after 9 DIV with GABAergic cells on the left and glutamatergic cells on the right. The lower panels show the cell-specific neuronal markers, GAD67 and Emx1 and the glial cell marker, GFAP. (**C)** Exemplary curves from the RT-qPCR amplification of BDNF. (**D)** Specific neuronal markers, GAD67 and Emx1, for GABAergic and glutamatergic neurons. (**E**) Relative quantification of BDNF, GAD67, Emx1 and GFAP in GABAergic cells and BDNF, GAD67 and Emx1 in hippocampal granule cells. (**F**) Relative quantification results for BDNF, GAD67 and Emx1 in GABAergic and glutamatergic neurons: FACS-isolated and cultured for 7–9 DIV and FACS-isolated and pelleted cell samples (pellet). Scale bar: (B), 100 µm.

**Table 1 ijms-21-01567-t001:** Overlap of mBDNF with GM130 and proBDNF with lamin signal detected in immunolabeled hippocampal YFP^+^ neurons.

Region	mBDNF + GM130 (%)	proBDNF + lamin (%)
DG (hilus)	11.0	48.1
CA3 (radiatum)	17.7	68.5
CA3 (oriens)	23.7	49.5
CA1 (radiatum)	40.7	43,.6
CA1 (oriens)	40.1	72.9
Average	26.6 ± 6.0 *	56.5 ± 5.9 *

* Values are the mean overlap (± SEM) of mBDNF and GM130, and proBDNF and lamin in YFP-positive neurons in the hippocampal regions/layers.

**Table 2 ijms-21-01567-t002:** Colocalization of the precursor proBDNF protein in YFP^+^ INs in immunolabeled hippocampal sections.

Animals	YFP^+^ Neuron Number	proBDNF and YFP^+^ Neuron Number	Percentage of YFP Neurons Containing proBDNF (%)
Animal I	142 *	137 *	96.5
Animal II	179 *	150 *	83.0
Total	321 *	287 *	89.4

* Values are the numbers of YFP^+^ and YFP/proBDNF^+^ neurons from the hippocampi from two animals.

**Table 3 ijms-21-01567-t003:** Delta-Cp values for transcripts of BDNF and neuronal markers in LCM-captured hippocampal neurons and neuropil samples.

	CA1 + CA3, Pyramidal Cell Layer	DG, Granule Cell Layer	CA1 + CA3, Stratum Oriens and Radiatum	CA1 + CA3 Neuropil
Mean ΔCp	Glutamatergic Neurons	Glutamatergic Neurons	GABAergic INs	Cell Body-Free
Exon-IX BDNF	0.690 ± 0.393 *	0.585 ± 0.049 *	0.120 ± 0.073 *	n.d.
Emx1	1.890 ± 0.946 *	0.027 ± 0.004 *	0.000 ± 0.008 *	n.d.
GAD67	0.080 ± 0.060 *	1.376 ± 0.169 *	3.350 ± 0.935 *	n.d.

* Values are the mean delta-Cp ± SEM of 6 repetitions (from 6 animals) for pyramidal, GABAergic and neuropil samples; and of 2 repetitions (from 2 animals) for the dentate granule cell samples; n.d., not detected.

**Table 4 ijms-21-01567-t004:** Delta-Cp values for transcripts of BDNF and neuronal markers in samples of the two main cortical neuron populations at different postnatal stages obtained by FACS.

	P0, 7–9 DIV Cultures		P14-P22 FACS	
Mean ΔCp	Glutamatergic Neurons	GABAergic Neurons	Glutamatergic Neurons	GABAergic Neurons
Exon IX BDNF	0.047 ± 0.013 *	0.009 ± 0.004 *	0.066 ± 0.024 *	0.020 ± 0.010 *
Emx1	0.007 ± 0.005 *	0 *	0.003 ± 0.000 *	0 *
GAD67	0.008 ± 0.005 *	1.239 ± 0.405 *	0.028 ± 0.000 *	0.136 ± 0.034 *

* Values are the mean delta-Cp ± SEM of 3 repetitions (6 animals) for glutamatergic and GABAergic fractions at P0 and P14-P22.

**Table 5 ijms-21-01567-t005:** List of primary Antibodies for immunocytochemistry.

Antibody	Supplier and Cat. No.	Host	Dilution	Immunogen
**BDNF Protein Forms**				
proBDNF	Merck MABN110	Mouse	1:1000	Precursor BDNF ~34 kDa GST-tagged recombinant protein corresponding to human pro-BDNF
mBDNF	Bioss bs-4989R	Rabbit	1:200	mature BDNF ~13 kDa keyhole limpet hemocyanin (KLH)-conjugated synthetic peptide derived from human BDNF
**GABAergic marker protein**				
GAD65/67	Merck ABN904	Rabbit	1:200	KLH-conjugated linear peptide corresponding to 14 amino acids from the C-terminal region of human glutamate decarboxylase 2 (GAD2)
**Glutamatergic somatic marker protein**				
Emx1	Abcam Ab224343	Rabbit	1:1000	Recombinant fragment corresponding to human Emx1 amino acids 76–162
Astrocytic marker				
GFAP	SySy 173011	Mouse	1:1000	Recombinant protein corresponding to amino acids 1–432 from human GFAP
**Perinuclear marker**				
Lamin-B1	Abcam ab16048	Rabbit	1:500	Synthetic peptide corresponding to mouse lamin-B1 amino acids 400–500 (internal sequence) conjugated to KLH
**Golgi marker**				
GM130	BD Bioscences 610823	Mouse	1:100	Rat GM130 amino acids 869–982

**Table 6 ijms-21-01567-t006:** List of primers used for qPCR and amplification conditions.

Gene	Primer Sequence		Efficiency
	Forward Primer (5’–3’)	Reverse Primer (5’–3’)	
Class III β-tubulin	gcgcatcagcgtatactacaa	catggttccaggttccaagt	1.93
BDNF exon IX	gcctttggagcctcctctac	gcggcatccaggtaatttt	1.95
GAD67	tgcggacatatgtgagaaataca	ttccgggacatgagcagt	1.898
Emx1	ctctccgagacgcaggtg	ctcagactccggcccttc	2.005
GFAP	tcgagatcgccacctacag	gtctgtacaggaatggtgatgc	2.001
**Real-time PCR conditions**			
Initial denaturation		95 °C—10’	
Cycling (x55)		95 °C—10’’ 60 °C—30’’ 72 °C—15’’	

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
