# Peer review of "BDNF Expression in Cortical GABAergic Interneurons"

_ijms, 2020, doi:10.3390/ijms21051567_

Round 1

Reviewer 1 Report

In the manuscript, the authors report that BDNF can also be synthesized in the principal cells and interneurons of mouse neocortex and hippocampus, although the amount is smaller than the BDNF synthesized in the glutamatergic neurons. It is a well-written paper with good organized experiments to verify the hypothesis, but there are some minor concerns as follow.

In this study, the authors showed the proBDNF and mBDNF immunoreactivities separately. Data sheet provided by the company do not inform that the antiserum detects only mBDNF, so please include detailed information of anti-mBDNF antiserum in the Materials and Methods section. In the 2.2.2 section, the authors described results with simple values (lines 152-155 and 159) without histograms or table. If the authors provide some histograms or tables it would be helpful to understand the results to readers. In the Introduction section, the abbreviation of interneuron, INs, comes at the first in the line 49, but interneuron appears again in the line 58. Please correct it. Please remove underline in the line 157. There are some typos in this manuscript. For example, 'this' and 'reflects' in the line 116 may change to 'these' and 'reflect'. In addition, 'prinicpal' in the line 384 and 'convegently' in the line 453 should change to 'principal' and 'convergently'. Please check typo throughout the manuscript.  

Author Response

In the manuscript, the authors report that BDNF can also be synthesized in the principal cells and interneurons of mouse neocortex and hippocampus, although the amount is smaller than the BDNF synthesized in the glutamatergic neurons. It is a well-written paper with good organized experiments to verify the hypothesis, but there are some minor concerns as follow.

We thank the Reviewer for their positive assessment of our manuscript.

(…) In this study, the authors showed the proBDNF and mBDNF immunoreactivities separately. Data sheet provided by the company do not inform that the antiserum detects only mBDNF, so please include detailed information of anti-mBDNF antiserum in the Materials and Methods section.

Indeed, the data sheet did not specify that this antibody detects only mature BDNF protein, however the company showed several validation experiments including immunocytochemistry and western blot analysis. Western blot shows only one band below 15 kDa, which corresponds to the molecular weight of the mature BDNF at ~13-14 kDa, suggesting that the antibody specifically recognizes the monomer form of mature BDNF. We included this information in the Material and Method Section 4.3.

(…) In the 2.2.2 section, the authors described results with simple values (lines 152-155 and 159) without histograms or table. If the authors provide some histograms or tables it would be helpful to understand the results to readers.

Thank you for pointing out this shortcoming. We have reformulated the text in the Results to better describe the findings of the quantification. We also included two new tables, as suggested, presenting the quantitative data on the localization proBDNF in YFP+ neurons (Table 1) and overlap of proBDNF and lamin, as well as mBDNF and GM130 in hippocampal neurons (Table 2).

(…) In the Introduction section, the abbreviation of interneuron, INs, comes at the first in the line 49, but interneuron appears again in the line 58. Please correct it.

Thank you for pointing this out. We have checked the use of abbreviations and corrected inconsistencies throughout the text.

(…) Please remove underline in the line 157. There are some typos in this manuscript. For example, 'this' and 'reflects' in the line 116 may change to 'these' and 'reflect'.5.) In addition, 'prinicpal' in the line 384 and 'convegently' in the line 453 should change to 'principal' and 'convergently'. Please check typo throughout the manuscript.

Thank you for flagging up these issues in the text. We have thoroughly checked the manuscript, removed the underline in line 157, and corrected the indicated typos, and thoroughly revised the text.

Reviewer 2 Report

In the paper by Tomas et al., the authors study the expression of BDNF in pyramidal cells and GABAergic interneurons in transgenic mice. Using tissue sections and cell cultures, they show that BDNF mRNA and protein can be detected in YFP-labeled GABAergic interneurons. BDNF gene products were detected in cell bodies, and not in cell body-free neuropil. In LCM experiments, control samples from the neuropil was found to lack BDNF expression, speaking against contamination of BDNF mRNA from other sources than cell bodies of principal cells and interneurons. Control experiments were done to test for glial sources.

Overall, the experiments are well carried out, and the results are clearly presented. The discussion is thorough. Importantly, the authors discuss that mature BDNF and proBDNF may be taken up from the extracellular space by endocytosis, leaving the possibility open that the finding of BDNF protein in a cell does not prove that BDNF is synthesized in it.

Some specific comment and suggestions are given below.

1.

The authors should explain what proportion of GABAergic neurons are labeled in VGAT-YFP mice. Are all GABAergic interneurons labeled, or just a fraction?

2.

Hippocampal interneurons are very diverse and thought to serve different functions. The authors state that they find mBDNF in interneurons across all hippocampal layers. This is not very well illustrated by the figures, and the authors could strengthen their conclusion by showing interneurons in the different layers positive for BDNF.

3. 

Line 158: 'The colocalization analyzed in the in dendritic layers of CA1 and CA3, where mostly INs are present indicated’. This sentence is not well written, and revision is needed.

4.

The author might want to discuss that proBDNF can control excitability of cortical pyramidal cells, as found by Gibon et al., J Neurosci. 2015, 35(26): 9741–9753.

5.

Conclusions, line 703: ‘In our present study we showed that cortical and hippocampal GABAergic neurons are generally able to synthesize BDNF under basal conditions’.

The authors have not proved that interneurons synthesize BDNF, but they have documented that some components necessary for synthesis may be present. Much more mechanistic experiments are needed to prove BDNF synthesis in GABAergic cells. The author should modify the text accordingly.

6.

The authors should discuss the implication of their findings in epilepsy.

Author Response

In the paper by Tomas et al., the authors study the expression of BDNF in pyramidal cells and GABAergic interneurons in transgenic mice. Using tissue sections and cell cultures, they show that BDNF mRNA and protein can be detected in YFP-labeled GABAergic interneurons. BDNF gene products were detected in cell bodies, and not in cell body-free neuropil. In LCM experiments, control samples from the neuropil was found to lack BDNF expression, speaking against contamination of BDNF mRNA from other sources than cell bodies of principal cells and interneurons. Control experiments were done to test for glial sources.

Overall, the experiments are well carried out, and the results are clearly presented. The discussion is thorough. Importantly, the authors discuss that mature BDNF and proBDNF may be taken up from the extracellular space by endocytosis, leaving the possibility open that the finding of BDNF protein in a cell does not prove that BDNF is synthesized in it.

Some specific comment and suggestions are given below.

We thank the Reviewer for this positive evaluation of our study and its presentation and the helpful comments.

1.) The authors should explain what proportion of GABAergic neurons are labeled in VGAT-YFP mice. Are all GABAergic interneurons labeled, or just a fraction?

We thank the Reviewer for this suggestion. Transgenic VGAT-YFP mice as been well characterized in the original publication of Wang et al. (Neuroscience 2009), but we have included the relevant information in the Results, section 2.1., line 90:

“In this mouse line 97.3 % of all hippocampal and 95.9 % of neocortical GABAergic neurons are labeled with YFP-Venus as described previously [27], and present a characteristic distribution throughout all hippocampal layers (Fig. 1 A, C, E, G).”

2.) Hippocampal interneurons are very diverse and thought to serve different functions. The authors state that they find mBDNF in interneurons across all hippocampal layers. This is not very well illustrated by the figures, and the authors could strengthen their conclusion by showing interneurons in the different layers positive for BDNF.

Thank you for pointing this out. Indeed, we only showed an overview of the hippocampus and an exemplarily an image of the hippocampal CA3. Following the reviewer's suggestion, we have prepared a new figure (Figure 4) showing examples of the localization for mBDNF and proBDNF in YFP+ neurons in the various layers of the main hippocampal regions (DG, CA3 and CA1).

Additionally, we have made a new figure (Figure 6) showing mBDNF and proBDNF labeling in purified GABA- and glutamatergic neuronal cultures.

We feel these results strengthen our conclusions that both mBDNF and proBDNF is expressed in the two major classes of cortical neurons both in situ and in vitro and that proBDNF preferentially localize to the rough endoplasmic reticulum in the perinuclear area, the main locus of protein synthesis.

3.) Line 158: 'The colocalization analyzed in the in dendritic layers of CA1 and CA3, where mostly INs are present indicated’. This sentence is not well written, and revision is needed.

We are sorry for the confusing formulation. We have reworded this sentence in the revised version (page 5):

“Next, we examined the signal overlap of the two proteins in the dendritic layers of CA1 and CA3, where many of the cell bodies of GABAergic INs are situated. This quantification showed that 56.5 ± 5.9 % of the proBDNF signal colocalized with lamin-B1 (Fig. 3 J; Table 2).”

4.) The author might want to discuss that proBDNF can control excitability of cortical pyramidal cells, as found by Gibon et al., J Neurosci. 2015, 35(26): 9741–9753.

Thank you for pointing out this important paper. We have added a sentence with a reference to Gibon et al. 2015 into the Discussion, Section (3.4.).

5.) Conclusions, line 703: ‘In our present study we showed that cortical and hippocampal GABAergic neurons are generally able to synthesize BDNF under basal conditions’.

The authors have not proved that interneurons synthesize BDNF, but they have documented that some components necessary for synthesis may be present. Much more mechanistic experiments are needed to prove BDNF synthesis in GABAergic cells. The author should modify the text accordingly.

We agree with the Reviewer’s rigorous assessment. We have changed our conclusions accordingly.

6.) The authors should discuss the implication of their findings in epilepsy.

We thank the Reviewer for this interesting suggestions, however, we think that our findings do not directly relate to epilepsy. Therefore, we feel reluctant to include a more detailed discussion about implications of our findings in epilepsy.

Nevertheless, in the context of the Gibson et al. 2015, we mention the potential relevance of proBDNF for excitability and epilepsy in a new sentence added to the Discussion (Section 3.4.).

Round 2

Reviewer 2 Report

The paper is greatly improved. 

In a few instances, especially in Section 4.7, proof reading is needed (spaces between text and brackets), i.e. 'captured cells, N=3(each of 2 animals', '(class III ß-Tubulin)in the'